# Interneuronal In Vivo Transfer of Synaptic Proteins

**DOI:** 10.3390/cells12040569

**Published:** 2023-02-10

**Authors:** Michael Klinkenberg, Michael Helwig, Rita Pinto-Costa, Angela Rollar, Raffaella Rusconi, Donato A. Di Monte, Ayse Ulusoy

**Affiliations:** German Center for Neurodegenerative Diseases (DZNE), 53127 Bonn, Germany

**Keywords:** protein spreading, oligomerization, animal models, Parkinson’s disease, vagus nerve

## Abstract

Neuron-to-neuron transfer of pathogenic α-synuclein species is a mechanism of likely relevance to Parkinson’s disease development. Experimentally, interneuronal α-synuclein spreading from the low brainstem toward higher brain regions can be reproduced by the administration of AAV vectors encoding for α-synuclein into the mouse vagus nerve. The aim of this study was to determine whether α-synuclein’s spreading ability is shared by other proteins. Given α-synuclein synaptic localization, experiments involved intravagal injections of AAVs encoding for other synaptic proteins, β-synuclein, VAMP2, or SNAP25. Administration of AAV-VAMP2 or AAV-SNAP25 caused robust transduction of either of the proteins in the dorsal medulla oblongata but was not followed by interneuronal VAMP2 or SNAP25 transfer and caudo-rostral spreading. In contrast, AAV-mediated β-synuclein overexpression triggered its spreading to more frontal brain regions. The aggregate formation was investigated as a potential mechanism involved in protein spreading, and consistent with this hypothesis, results showed that overexpression of β-synuclein, but not VAMP2 or SNAP25, in the dorsal medulla oblongata was associated with pronounced protein aggregation. Data indicate that interneuronal protein transfer is not a mere consequence of increased expression or synaptic localization. It is rather promoted by structural/functional characteristics of synuclein proteins that likely include their tendency to form aggregate species.

## 1. Introduction

A significant feature of Parkinson’s disease (PD) is the progressive accumulation of α-synuclein (α-syn)-containing inclusions in the brain and peripheral tissues [1,2]. Neuropathological studies and experimental evidence support the possibility that, in PD and PD models, α-syn pathology may spread from the site(s) of its initial accumulation to other interconnected sites following a stereotypical pattern [3,4]. Mechanisms of this spreading and its specificity to the α-syn protein remain relatively unclear, however. Investigations into specific features of α-syn that may contribute to its spreading could, therefore, provide critical clues on the molecular mechanisms underlying this phenomenon.

α-Syn is a natively unfolded protein enriched in pre-synaptic nerve terminals. Specific localization of α-syn in the synapse is mediated, at least in part, by its ability to bind to lipids (i.e., synaptic membranes) and pre-synaptic proteins [5,6]. From the pathological standpoint, findings of genetic studies indicate that changes in α-syn amino acid sequence due to missense mutations or increased α-syn expression caused by *SNCA* multiplication mutations trigger genetic forms of parkinsonism [7,8,9,10,11,12,13,14,15,16]. Interestingly, these disease-causing modifications are also associated with an increased propensity to aggregate and altered vesicular binding, suggesting that protein levels and amino acid sequence that affect α-syn’s ability to aggregate or to bind to other molecules may play an important role in PD pathogenetic processes [6,17,18].

In the brain, pathological α-syn accumulation in the form of Lewy inclusions is not randomly distributed but targets specific neuronal populations. Cholinergic neurons in the dorsal motor nucleus of the vagus nerve (DMnX) represent one of the primary sites of α-syn burden at the early stages of disease development [19]. As the disease progresses, the pathology often advances from this nucleus in the lower brainstem toward specific neuronal populations in more rostral brain regions [19,20]. The stereotypical pattern of this spreading supports the hypothesis that progressive pathology may arise from the interneuronal transfer of pathogenic α-syn species between anatomically connected brain regions. A variety of in vitro and in vivo experimental models have been developed and utilized to investigate this hypothesis [1,20,21]. For example, valuable clues on interneuronal α-syn spreading involving the DMnX have been obtained using an in vivo paradigm consisting of unilateral injection of adeno-associated viral vectors (AAVs) designed to express human α-syn (AAV-α-syn) into the mouse or rat vagus nerve [22,23,24,25,26]. This treatment first induces overexpression of human α-syn targeted to the dorsal medulla oblongata (dMO). It is then followed by neuron-to-neuron transfer and caudo-rostral advancement of human α-syn that progressively and stereotypically spreads towards pontine, midbrain, and forebrain regions [22,24,26,27]. Important findings obtained in this model include the demonstration that aggregated forms of α-syn and, in particular, oligomeric protein species, are likely to play an important role in neuron-to-neuron α-syn transfer [24].

A key question raised by the finding of overexpression-induced α-syn spreading concerns the specificity of this effect and the extent to which specific α-syn properties may underlie it. To address this question, experiments were carried out in animals that were intravagally injected with AAV vectors delivering green fluorescent protein DNA (AAV-GFP). Similar to the results after treatment with AAV-α-syn, these animals showed robust protein (GFP instead of human α-syn) expression in the dMO; this overexpression did not result, however, in caudo-rostral GFP spreading [22,24,26,27]. Lack of protein spreading in AAV-GFP-treated animals provided initial evidence suggesting that interneuronal spreading is not a mere consequence of protein overexpression. These experiments, however, did not thoroughly address the question of the specificity of the interneuronal α-syn transfer. It could be argued, for example, that GFP is a cytosolic (as compared to synaptic) protein almost double in size as compared to α-syn. Therefore, if the size and subcellular localization of a protein are important factors determining its neuron-to-neuron exchange, lack of GFP transfer would not itself rule out the possibility that the spreading properties of α-syn may be shared by other proteins.

The present study was designed to further investigate the specificity of overexpression-induced α-syn spreading. To achieve this goal, the spreading potential of proteins similar to α-syn in size, amino acid sequence, and subcellular localization (i.e., pre-synaptic) was assessed after induction of their overexpression in the mouse medulla oblongata. Experiments were also carried out to test the hypothesis that a relationship exists between interneuronal protein transfer and the formation of protein aggregates. The proteins investigated were β-syn, vesicle-associated membrane protein 2 (VAMP2), and synaptosomal-associated protein of 25kDa (SNAP25). β-Syn is a synuclein family protein with size and amino acid sequence very similar to α-syn. β-syn is not only expressed within the same neuronal populations in the brain but also shares with α-syn its synaptic localization [28]. VAMP2 and SNAP25 are key components of the soluble N-ethylmaleimide-sensitive fusion attachment protein receptor (SNARE) complex known to interact with α-syn in the synaptic compartment [29,30]. Results revealed marked differences in spreading capabilities among different proteins, supporting specific interneuronal mobility of synuclein proteins. This mobility could be mediated, at least in part, by the propensity of α- and β-syn to form aggregates within overexpressing neurons.

## 2. Materials and Methods

### 2.1. Viral Vectors

Serotype 6 recombinant AAVs were used for transgene expression of β-syn and human influenza hemagglutinin (HA)-tagged β-syn, VAMP2, SNAP25, and α-syn under the control of a human synapsin 1 promoter. All transgenes were human-derived and wild-type. Gene expression was enhanced using a woodchuck hepatitis virus posttranscriptional regulatory element (WPRE) and a polyA signal. AAV vector production, purification, concentration, and titration were performed by Vector Biolabs (Philadelphia, PA, USA). Vectors were diluted in phosphate-buffered saline solution to titers ranging between 3.5–5.0 × 10^12^ genome copies/mL in order to achieve similar transduction in vivo.

### 2.2. Animals and Surgical Procedures

Animal experiments were approved by the State Agency for Nature, Environment, and Consumer Protection in North-Rhine Westphalia, Germany. Experiments were conducted in female C57BL/6JRj mice (Janvier Labs, Le Genest-Saint-Isle, France) between 15 and 22 weeks of age. Animals were housed in individually ventilated cages in a specific-pathogen-free facility and kept on a 12-hour light/dark cycle with ad libitum access to food and water. To induce transgene expression, AAV vectors were injected into the left vagus nerve as previously described [24]. All surgical procedures were performed under isoflurane anesthesia and buprenorphine analgesia. A small incision was made at the midline of the neck, and the left vagus nerve was isolated. The AAV-containing solution (800 nL) was injected at a flow rate of 350 nL/min using a 35-gauge blunt steel needle fitted onto a 10 μL NanoFil syringe. Mice were sacrificed 6 weeks after the surgery with a lethal dose of sodium pentobarbital (i.p.) and perfused through the ascending aorta with 4% (*w*/*v*) paraformaldehyde. Brains were removed and immersed in 4% paraformaldehyde solution for 24 h before cryopreservation in 30% (*w*/*v*) sucrose.

### 2.3. Tissue Preparation and Immunohistochemistry

Coronal brain sections (35 μm) were generated using a freezing microtome. For brightfield microscopy, free-floating sections were quenched by incubation in a mixture of 3% H_2_O_2_ and 10% methanol in Tris-buffered saline (pH 7.6). Non-specific binding sites were blocked using 5% normal serum. Samples were kept overnight at room temperature in a solution containing the primary antibody: rabbit anti-β-syn (1:5000; ab6165, Abcam, Cambridge, UK), rabbit anti-HA (1:15,000; clone C29F4, Cell Signaling, Danvers, MA, USA), or rabbit anti-human α-syn (1:50,000; clone MJFR1, Abcam, Cambridge, UK). Sections were then rinsed and incubated in a biotinylated secondary antibody solution (1:200; Vector Laboratories, Newark, CA, USA). Following treatment with avidin–biotin–horseradish peroxidase complex (ABC Elite kit, Vector Laboratories, Newark, CA, USA), a color reaction was developed using a 3,3′-diaminobenzidine kit (Vector Laboratories, Newark, CA, USA). Sections were mounted on coated slides and coverslipped with Depex (Sigma-Aldrich, St. Louise, MO, USA). For fluorescent labeling, heat-induced epitope retrieval was carried out on free-floating medulla and pons sections with sodium citrate buffer (10 mM plus 0.05% Tween, pH 6.0) for 5 min at 95 °C. Samples were then blocked with 5% normal serum and incubated with rabbit anti-β-syn polyclonal antibody (1:1000; ab6165, Abcam, Cambridge, UK) and mouse anti-HA antibody (1:1000; clone HA-7, Sigma-Aldrich, St. Louise, MO, USA ) overnight at 4 °C. Labeling was performed with anti-rabbit Dylight 488 and anti-mouse DyLight 594 (1:300; Vector Laboratories, Newark, CA, USA) antibodies. Sections were mounted on object slides and coverslipped with ProLong^™^ Gold Antifade Mountant (Invitrogen, Waltham, MA, USA).

### 2.4. In situ Proximity Ligation Assay (PLA)

Free-floating sections of the medulla oblongata were processed using Duolink (Sigma-Aldrich, St. Louise, MO, USA) according to the manufacturer’s protocols as described previously [24,25,26]. HA and α-syn PLA probes were generated by linking the plus and minus oligonucleotides with primary antibodies anti-HA (clone C29F4, Cell Signaling) and anti-human α-syn (clone syn211, Millipore, Burlington, MA, USA) using a probemaker kit (Duolink, Sigma-Aldrich, Janvier Labs). HA/HA and α-syn/α-syn interactions were detected using a direct PLA method, which required incubating sections overnight in solutions containing the plus and minus PLA probes directly conjugated to the HA (1:250) or human α-syn–antibody (1:100; mouse-anti-Syn211, Millipore, Burlington, MA, USA). Samples treated only with the plus probe served as negative controls. Following ligation and amplification, specific PLA signals were visualized using a brightfield detection kit (Duolink, Sigma-Aldrich, St. Louise, MO, USA). Sections were mounted on object slides and coverslipped using histomount mounting media (Life Technologies, Carlsbad, CA, USA).

### 2.5. RT-PCR

Brain tissue from 5 mice per treatment group was used to assess mRNA expression. Extraction of total RNA and analysis of mRNA expression by RT-PCR was carried out as previously described [24]. All viral vectors used in this study contained a WPRE sequence. Comparison of the expression levels of different transcripts, i.e., β-syn, VAMP2, and SNAP25, was therefore carried out by measuring WPRE mRNA levels. Target-specific primer pairs were generated for the WPRE (forward 5′ CAATTCCGTGGTGTTGTCGG and reverse 5′ CAAAGGGAGATCCGACTCGT) and the housekeeping gene hypoxanthine guanine phosphoribosyltransferase (Hprt) (forward: 5′ TCCTCCTCAGACCGCTTTT and reverse: 5′ CCTGGTTCATCATCGCTAATC. RT-PCR-amplified WPRE and Hprt products were run in prestained (Red safe, Intron Biotechnology, Seongnam-Si, Republic of Korea) agarose gels. Images were scanned with an InGenius3 imaging system, acquired with GeneSys software (v.1.5.7.0), and quantified using GeneTools analysis software (v.4.3.14.0, Syngene, Bangalore, India). The ratio of blank corrected raw volume values of WPRE and Hprt were calculated and expressed as WPRE/Hprt ratio.

### 2.6. Thioflavin-S Staining

Four mice from each experimental group were used for Thioflavin-S labeling. Medulla oblongata sections containing the DMnX were first stained with anti-HA and labeled using a secondary antibody conjugated with Dylight 594, as described above. Stained sections were mounted on glass slides and were incubated with 0.05% Thioflavin-S (Sigma-Aldrich, St. Louise, MO, USA) for 8 min, followed by sequential differentiation in 80–95–95% ethanol (3 min each). Sections were briefly rinsed in water, coverslipped with Prolong Gold antifade reagent (Thermo Fischer, Waltham, MA, USA), and the images were acquired on the same day.

### 2.7. Image Acquisition

Brightfield tile scan images of the dorsal medulla oblongata were acquired using an Observer.Z1 microscope (Carl Zeiss, Oberkochen, Germany) equipped with a motorized stage using a Plan-Apochromat 20×/0.8 M27 objective (Carl Zeiss, Oberkochen, Germany). Raw images were corrected for camera shading, and tiles were stitched with ZEN 2012 software (Blue edition, Carl Zeiss, Oberkochen, Germany). Confocal stack images were taken with a Plan-Apochromat 63×/1.4 oil objective on a Zeiss LSM800 and illustrated as maximum-intensity projection images. PLA images and high magnification brightfield stack images were generated on an Olympus BX53 widefield light microscope using a 10× Plan-Apochromat and 100×/1.40 UPlanSApo oil objective, respectively. Stack images were processed as minimum intensity projections with Stereoinvestigator^®^ 10 software (MBF Biosciences, Williston, VT, USA). For image analysis purposes, sections assayed with PLA were scanned using an automated upright slide scanning microscope (AxioScan.Z, Carl Zeiss, Oberkochen, Germany).

### 2.8. Axonal Counts and Image Analysis

HA-immunoreactive axons were counted by a blinded investigator. *N* ≥ 7 mice /group were used for these analyses. Sections were analyzed from caudal to rostral in the left hemisphere at pre-defined bregma coordinates in the pons (bregma −5.40 mm), the midbrain (bregma −4.60 mm), and the forebrain (bregma −1.70 mm) with an Axio Scope.A1 brightfield microscope (Carl Zeiss, Oberkochen, Germany) using EC Plan-Neofluar 40× oil objective. HA staining intensity, PLA signal intensity, and area occupied by the PLA dots were assessed using Fiji (Image J, version 2.1.0/1.53c). Images collected from HA immunostained MO (*n* = 5 mice/group) and tissue sections processed from PLA (*n* ≥ 4 mice/group) were converted to grayscale images and inverted. Background subtraction using a rolling ball radius of 50 pixels was applied. The left (injected side) dorsal vagal complex encompassing the DMnX, nucleus of tractus solitarius, and area postrema was delineated. HA staining intensity was then measured within this delineated area. For PLA image analyses, a fixed intensity threshold was set to automatically outline the PLA dots, and the PLA dots were analyzed using the “analyze particles” function of ImageJ (v.2.1.0/1.3Jc). The intensity of the PLA signal and the percent of area occupied by the PLA dots were measured for each animal.

### 2.9. Statistical Analyses

Prism 9 (v9.1.0, GraphPad Software, Boston, MA, USA) was used for all statistical analyses. Prior to using parametric tests, the normal distribution of the data was confirmed using the Shapiro–Wilk test for normality. A 2-tailed unpaired t-test was used to compare 2 groups, and a one-way ANOVA followed by Tukey’s posthoc test was used to compare three groups. Data were expressed as mean ± standard error of the mean. *p*-Values less than 0.05 were considered statistically significant.

## 3. Results

### 3.1. Effects of β-syn Transduction

To assess whether size, amino acid sequence, and/or synaptic localization contribute to interneuronal protein transfer, we first assessed the spreading potential of a protein highly similar to α-syn, namely β-syn (Figure 1a). For these experiments, mice were treated using a paradigm capable of inducing targeted protein overexpression in the DMnX-containing dMO [22,24]. They received a unilateral injection of AAVs mediating human β-syn expression (AAV-β-syn) into the left vagus nerve and were kept for 6 weeks after surgery (Figure 1b). At this time point, brains were processed for histological assessment of β-syn in coronal sections of the medulla oblongata and pons that were stained with a β-syn antibody (Figure 1c–e). The purpose of these analyses was to detect levels of expression in the dMO and potential spreading (in the pons) of the exogenous protein, i.e., human β-syn. It is noteworthy that β-syn amino acid sequence is highly conserved and features 97% homology between humans and mice [31]. Therefore, antibodies against β-syn, including the reagent used for this study, are not species-specific and would react against both the human and murine protein. To circumvent this limitation, staining of our tissue sections was carried out using a relatively high antibody dilution that allowed us to reduce the basic immunoreactivity due to endogenous β-syn and to detect the effects of AAV-induced overexpression more distinctly. Targeted transduction was indicated by robust β-syn immunoreactivity in the dMO (Figure 1f).

The pattern of overexpression was consistent with the anatomical distribution of vagus-associated neurons and with the results of earlier studies that used the intravagal route of AAV administration for medullary protein overexpression [22,24]. In particular, AAV-mediated β-syn transduction affected cell bodies and neurites in the left DMnX (ipsilateral to the AAV injection) (Figure 1f,g); it was also evident within afferent vagal projections in the left and, to a lesser extent, right nucleus of the tractus solitarius and in the area postrema (Figure 1f,h). Next, coronal sections of the pons were immunostained for β-syn since the detection of β-syn-loaded axons in this brain region would indicate its interneuronal transfer [22,24]. Analyses revealed that strongly labeled axons were scattered throughout pontine sections in AAV-injected mice but not in control specimens from non-injected animals (Figure 1i,j). Taken together, these data suggested that, following its overexpression in the dMO, β-syn was able to spread toward more rostral brain regions and became accumulated within pontine axons. Important limitations of these analyses included the lack of specific detection of endogenous (mouse) vs. exogenous (human) β-syn that prevented a more refined characterization of protein overexpression in the dMO and a clearer visualization and quantification of the spreading protein in pontine as well as more rostral brain regions.

### 3.2. Overexpression and Spreading of HA-Tagged α-syn

To overcome the technical issue described above, new AAVs were generated that encoded α-syn with an HA tag at its C-terminal end (AAV-β-syn-HA) (Figure 2a). Mice received a single injection of this AAV-β-syn-HA into the left vagus nerve, and 6 weeks later, their brains were processed for histological assessment. For the first set of analyses, coronal tissue sections of the medulla oblongata were immunostained using antibodies raised to detect β-syn or HA. Microscopic images of β-syn immunoreactivity were similar in samples from mice injected with AAV-β-syn-HA as compared to animals treated with AAV-β-syn (cf. Figure 1f and Figure 2b); they showed increased labeling of dMO neurons on a relatively high background staining (Figure 2b). This background staining was abolished when specimens from AAV-β-syn-HA-injected mice were labeled with anti-HA, allowing a clear, unambiguous detection of transgenic human β-syn within vagus-associated dMO neurons (Figure 2c). Next, medullary sections from AAV-β-syn-HA-injected mice were double-labeled with anti-β-syn and anti-HA and processed for immunofluorescence. Results showed consistent co-localization, thus confirming a precise detection of β-syn using the HA antibody (Figure 2d). In particular, double-labeled neuronal cell bodies were evident in the DMnX, indicating neuronal transduction and overexpression (Figure 2d, higher magnification). Anti-β-syn and anti-HA were also used to double-label pontine sections and assess this tissue for the presence of extra-medullary human β-syn. Distinct, co-labeled axons were detected as a result of this analysis, pointing to a caudo-rostral (medullary-to-pontine) spreading of the “entire” (human β-syn plus HA) exogenous protein (Figure 2e).

Having demonstrated the suitability of staining with anti-HA as a means to detect exogenous β-syn in AAV-β-syn-HA-injected mice, a detailed evaluation of protein spreading was performed in coronal sections throughout the brain of these animals that were stained with anti-HA and analyzed with brightfield microscopy. Enlarged, dystrophic axons containing β-syn were observed in all sections of the pons, midbrain, and forebrain that were collected at predefined Bregma coordinates (Figure 2f). Consistent with the results of earlier reports, spreading predominantly affected the side of the brain ipsilateral to the AAV injection and occupied preferential sites; these sites included the locus coeruleus in the pons, the reticular formation (but not the substantia nigra) in the midbrain, and the central nucleus of the amygdala in the basal forebrain [22,23,24,26,32,33]. For quantification purposes, the number of HA-immunoreactive axons was counted and found to be higher in sections closer to the medulla oblongata, i.e., in the pons, and progressively lower in other more rostral brain regions (Figure 2g). The number of axons containing exogenous β-syn was consistently 60–70% lower on the contralateral as compared to the ipsilateral side of the brain. In particular, when bilateral counts were carried out in the pons of 5 AAV-injected mice, the number of β-syn-labeled axons was found to be 32.6 ± 6.1 and 87.6 ± 12.6 contralaterally and ipsilaterally, respectively.

### 3.3. Lack of Interneuronal Spreading of VAMP2 and SNAP25

The next set of experiments was designed to test the spreading potential of VAMP2 and SNAP25, two proteins that share their synaptic localization with α- and β-syn. To induce overexpression, mice were injected intravagally with AAVs encoding HA-tagged proteins, namely VAMP2-HA and SNAP25-HA (Figure 3a). The rationale for overexpressing tagged proteins was twofold. First, this strategy was chosen in order to improve the detection of the transduced protein, as indicated by the results of experiments using tagged β-syn (see above). Secondly, overexpression of tagged proteins allowed us to standardize our comparative analyses since the same antibody (anti-HA) could be used for quantification of protein spreading after transduction with β-syn, VAMP2, or SNAP25.

Mice were sacrificed, and brains were collected and sectioned at 6 weeks after treatment with AAV-VAMP2-HA or AAV-SNAP25-HA. To verify and compare gene expression, tissue specimens of the left (ipsilateral to the AAV injection side) dMO were collected from mice injected with AAV-VAMP2-HA or AAV-SNAP25-HA as well as from animals treated with AAV-β-syn-HA. They were then processed for RT-PCR and assayed for the presence of WPRE (an enhancer element contained in all our vector constructs) and Hprt (used as a housekeeping gene) sequences. A 204 bp WPRE band and a 98 bp Hprt band were present in all samples from AAV-injected animals (Figure 3b). Semi-quantification of band intensities revealed similar levels of WPRE mRNA, indicating that comparable transduction was achieved with each of the three different vectors (Figure 3c). HA immunohistochemistry on medullary tissue sections from mice treated with AAV-VAMP2-HA or AAV-SNAP25-HA showed clear and robust expression of both transgenes in the dMO (Figure 3d). The expression pattern of VAMP-2-HA or SNAP25-HA was similar to the pattern of overexpression of β-syn-HA previously described in animals treated with AAV-β-syn-HA (cf. Figure 2c and Figure 3d); it included a robust protein accumulation within neuronal cell bodies in the left DMnX (Figure 3e). HA staining intensity was also comparable in the left dMO of mice injected with AAV-VAMP2-HA, AAV-SNAP25-HA, or AAV-β-syn-HA, further confirming that similar transduction and overexpression levels were achieved after treatment with three different vectors (Figure 3f).

Protein spreading was assessed and quantified by counting the number of HA-immunolabeled axons in predefined sections of the pons, midbrain, and forebrain. All brain sections from mice injected with AAV-VAMP2-HA were consistently devoid of HA immunoreactivity, whereas only a few very scant HA-labeled axons were seen in pontine (but not midbrain or forebrain) sections from SNAP25-HA-expressing animals. A comparison of these data with findings that were obtained after β-syn-HA overexpression revealed clear differences and indicated that, among these three synaptic proteins (VAMP2, SNAP25, and β-syn), only β-syn possessed overt neuron-to-neuron spreading capability (Table 1).

### 3.4. Detection and Assessment of Aggregation of HA-Tagged Proteins

Results of earlier reports investigating the spreading properties of α-syn suggest that the formation of protein aggregates could promote interneuronal protein transfer [24,34,35,36,37]. In particular, soluble oligomeric species, which were detected by a proximity ligation assay (α-syn/α-syn PLA), were proposed as possible mediators of human α-syn transfer after its AAV-induced overexpression in the mouse dMO [24,25,26]. Based on these earlier data, a set of experiments was designed here to test the hypothesis that differences in spreading among synaptic proteins may be due, at least in part, to their different ability to form aggregate species.

To detect and compare aggregate formation after AAV-induced overexpression of HA-tagged proteins, a new PLA was developed; the assay was designed to recognize HA molecules in close proximity as markers of protein assembly. An important initial step in the development and validation of this HA/HA PLA was to carry out an experiment in mice that were injected intravagally with AAV vectors encoding for HA-tagged human α-syn (AAV-α-syn-HA) (Figure 4a). Since overexpression-induced aggregation of human α-syn has already been demonstrated using α-syn/α-syn PLA [24], the aim of these experiments was to analyze samples using either α-syn/α-syn or HA/HA PLAs and to determine if the two assays yielded comparable results. In mice injected with AAV-α-syn-HA and sacrificed 6 weeks later, robust protein overexpression was seen in the dMO after staining with an antibody raised against human α-syn; a similar pattern and intensity of immunostaining were observed in tissue sections labeled anti-HA (Figure 4b). Medullary sections from AAV-α-syn-HA-treated animals were then processed for α-syn/α-syn or HA/HA PLA. Results using the two complementary (negative and positive) α-syn/α-syn or HA/HA PLA probes showed clear labeling in vagus-associated medullary neurons, including cells in the DMnX (Figure 4c). The specificity of this labeling was confirmed by its absence in control samples that were processed without the negative PLA probe (Figure 4c). Further semi-quantitative assessment of PLA reactions was carried out by measuring PLA signal intensity and by calculating the dMO area occupied by the PLA signal. Comparative analyses indicated that similar signal intensities and similar areas of signal distribution characterized samples that were processed either with α-syn/α-syn or HA/HA PLA (Figure 4d). Data therefore demonstrated the suitability and effectiveness of HA/HA PLA as a technique capable of detecting the aggregation of HA-tagged proteins.

HA/HA PLA was then used to investigate protein assembly in mice injected with AAV-β-syn-HA, AAV-VAMP2-HA, or AAV-SNAP25-HA. Analyses indicated clear differences between the first vs. the other two groups of animals. Specific PLA signals were indeed detected in animals overexpressing β-syn-HA, whereas medullary sections from mice overexpressing VAMP2- or SNAP25-HA were consistently devoid of labeling (Figure 5a,b). Further evidence of β-syn-HA aggregation was obtained from the analysis of medullary sections from AAV-β-syn-HA-injected mice stained with Thioflavin-S, a fluorescent dye that binds to amyloid structures. As shown in Figure 5c, Thioflavin-S labeling was detected within DMnX cell bodies and neurites that overexpressed β-syn-HA and were therefore co-stained with anti-HA. Consistent with the results of HA/HA PLA analyses, medullary sections from mice treated with AAV-VAMP2-HA or AAV-SNAP25-HA were devoid of Thioflavin/HA co-localization (Figure 5c). These data revealed the ability of β-syn, but not VAMP2 or SNAP25, to form aggregate species after its overexpression within vagus-associated neurons in the dMO. Together with the results on overexpression-induced protein spreading, these findings also support a relationship between the formation of protein aggregates and interneuronal β-syn transfer.

## 4. Discussion

The intravagal route of administration of viral vectors carrying specific coding DNA sequences has been successfully used to induce targeted protein overexpression in the dMO [22,23,24,25,26,27]. Earlier work has also demonstrated that, through this paradigm, AAV transduction and protein overexpression should remain confined within vagus-associated neurons. This anatomical restriction was evaded, however, when animals were treated intravagally with AAVs delivering human α-syn DNA. In these animals, the detection of human α-syn first in the pons and then in more rostral brain regions indicated a neuron-to-neuron transfer of the exogenous protein that resulted in its progressive spreading throughout the brain [22,23,24,25,26,27]. Caudo-rostral human α-syn advancement followed a stereotypical pattern and affected brain regions in the pons, midbrain, and forebrain that are anatomically connected to the dMO. The substantia nigra was not among these regions, probably because of its weak direct connections to the DMnX [22,38]. The present study aimed at investigating the interneuronal mobility of proteins other than human α-syn, addressing important questions: Is overexpression in the dMO sufficient to trigger protein spreading? Are synaptic proteins particularly prone to neuron-to-neuron transfer? In addition, do specific protein conformations confer greater protein mobility?

An initial key component of the study was to develop and validate tools and experimental paradigms most suitable to address these questions. The first technical obstacle to overcome related to the fact that basic endogenous protein expression prevented an accurate assessment, quantification, and comparison of the overexpression and potential spreading of the three proteins tested in this study, namely β-syn, VAMP2, or SNAP25. This limitation was successfully overcome by designing and using viral vectors that encoded β-syn, VAMP2, or SNAP25 with an HA tag at their C-terminal ends. Intravagal injections of these vectors induced expression of the tagged proteins that could all (β-syn-, VAMP2- or SNAP25-HA) be readily detected using the same HA antibody. Results of these experiments revealed similar levels of AAV-induced transduction and similar patterns of β-syn-HA, VAMP2-HA, or SNAP25-HA protein overexpression in the dMO. Quite in contrast, analyses of pontine, midbrain, and forebrain sections for the detection of extra-medullary proteins showed marked differences between mice transduced with AAV-β-syn-HA vs. animals injected with either AAV-VAMP2-HA or AAV-SNAP25-HA. In particular, a significant number of β-syn-HA-containing axons was observed in the pons, midbrain, and forebrain of AAV-β-syn-HA-treated animals, whereas transduction with AAV-VAMP2-HA or AAV-SNAP25-HA resulted in negligible or no detection of the corresponding protein in sections rostral to the medulla oblongata. It is noteworthy that cleavage and degradation of the HA tag, which have been reported by earlier investigations [39], might have hampered our assessment of protein overexpression (in the dMO) and protein spreading (in regions rostral to the medulla oblongata) using HA immunoreactivity. HA cleavage and clearance cannot be completely ruled out under our experimental conditions. Nonetheless, our findings do unequivocally indicate accumulation of β-syn-HA in the dMO and caudo-rostral spreading of β-syn-HA since co-labeling analyses consistently detected co-localization of HA and β-syn immunoreactivities.

Our results indicate that overexpression of a protein in the dMO does not necessarily trigger its interneuronal transfer. Similarly, protein size may not be a key factor in determining interneuronal transfer since two proteins with similar molecular weight, namely VAMP2 (13 kDa) and β-syn (14 kDa), displayed very different spreading abilities. Data also suggest that localization within neuronal synapses is not directly linked to protein transfer and is not sufficient to predict the spreading potential of a protein. Indeed, overexpression of α- and β-syn triggered their caudo-rostral spreading, whereas overexpression of VAMP2 or SNAP25, two other synaptic proteins, did not result in their detection and accumulation in brain regions rostral to the medulla oblongata. These marked differences in spreading ability may reflect important features of these synaptic proteins, such as their relative solubility (synucleins) and their ability to stably interact with synaptic membranes (VAMP2 or SNAP25), that could facilitate or impede their spreading potential. Further studies are warranted to address this hypothesis. On the other hand, taken together, our current findings support the conclusion that similarities in amino acid protein sequence between α- and β-syn are likely responsible for their interneuronal mobility and, ultimately, their ability to spread between interconnected brain regions. Sequence and structural similarities between the two synucleins could underlie shared properties, such as their ability to form aggregates (see below), directly relevant to their spreading behavior.

Results of previous studies support the conclusion that interneuronal α-syn transfer is promoted by the formation of aggregated forms of the protein [24,34,35,36]. This effect may be related to the ability of aggregated α-syn to interact specifically with proteins involved in the exchange of small molecules between cells [36]. Amino acid sequences can play an important role in determining the ability of proteins to form aggregates. We therefore hypothesized that, due to its similarity to α-syn, β-syn may also aggregate after its overexpression within medullary neurons; this aggregation may then underlie β-syn’s property to pass from neuron to neuron. As a corollary to this hypothesis, no significant aggregation would be expected to occur after overexpression of VAMP2 or SNAP25 since both these proteins were shown not to spread outside the dMO. Several conformation-specific antibodies are available to detect α-syn aggregates and have been used to distinguish between monomeric and aggregated forms of the protein [40,41,42,43]. No such in situ tools exist for assessing β-syn, VAMP2, or SNAP25 aggregates, and for this reason, another technical challenge of this study was to develop analytical assays capable of detecting β-syn, VAMP2, and SNAP25 assemblies. To achieve this goal, a new in situ PLA was established. PLA is a sensitive detection method that utilizes single-stranded oligonucleotides coupled with antibodies. Quite importantly, earlier studies have validated and used specific PLAs for detecting aggregated α-syn [24,44,45] as well as aggregated forms of other pathogenic proteins [46,47] in tissue from humans and animal models. Previous studies focusing on α-syn aggregation used a human-specific α-syn antibody and verified that this PLA method specifically detected aggregated rather than monomeric α-syn; results of these earlier investigations also indicated that the α-syn/α-syn PLA preferentially recognized oligomeric α-syn species [24,44,45]. Our current strategy was to develop, validate and use a PLA assay targeting the HA tags of α-syn-, β-syn-, VAMP2-, or SNAP25-HA proteins. In the dMO of mice overexpressing human α-syn-HA, this HA/HA PLA generated a specific signal that was comparable in intensity and distribution to the labeling obtained by processing the same tissues with α-syn/α-syn PLA. Following these validation experiments, HA/HA PLA was used to assess medullary sections from animals injected with AAV-β-syn-HA, AAV-VAMP2-HA, or AAV-SNAP25-HA. Results revealed a strong signal in the dMO of mice overexpressing β-syn-HA, whereas medullary tissues from mice overexpressing VAMP2-HA or SNAP25-HA were completely devoid of HA/HA PLA labeling. Further analyses using the fluorescent dye Thioflavin-S confirmed the accumulation of amyloid fibrils in specimens from animals treated with AAV-β-syn-HA but not AAV-VAMP2-HA or AAV-SNAP25-HA. Taken together, these results demonstrated the overexpression-induced formation of β-syn aggregates. They also indicated that spreading occurred only under conditions associated with protein (α- or β-syn) assembly, thus supporting a relationship between aggregation and interneuronal protein transfer.

Our current evidence showing aggregation and spreading of β-syn raises a few important considerations. Previous studies investigating the toxicity and pathogenicity of β-syn have reported findings that are apparently conflicting. Some studies concluded that β-syn, unlike α-syn, has a low propensity to aggregate and induce pathology and, in fact, may even possess protective properties and inhibit α-syn aggregation [48,49,50,51,52,53,54,55]. Other investigations reported that when overexpressed within nigral dopaminergic neurons, β-syn forms proteinase K-resistant aggregates and induces nigral neurodegeneration in rodents [56,57,58]. A likely explanation for these inconsistent data is that, while β-syn may not itself be an aggregation-prone protein, its assembly may be triggered by external factors. In support of this conclusion, earlier work showed that β-syn is capable of aggregating and becoming neurotoxic under a variety of conditions that include oxidative stress, presence of dopamine, and exposure to environmental toxins, metals, glycosaminoglycans, or macromolecular-crowding agents [57,58,59]. Based on these earlier findings, it is reasonable to infer that, in our present model of AAV-mediated β-syn overexpression, one or more of these factors could underlie protein aggregation and interneuronal protein transfer. For example, in mice with overexpression of α-syn in the dMO, an intriguing relationship has recently been reported between α-syn overexpression, oxidative stress, protein aggregation, and spreading [26]. Further studies are warranted to assess whether a similar relationship exists as a result of increased β-syn load. Results of these future studies could not only underscore the important role of oxidative stress in the development of proteinopathies but could also elucidate important mechanisms that may underlie β-syn pathogenicity.

## Figures and Tables

**Figure 1 cells-12-00569-f001:**
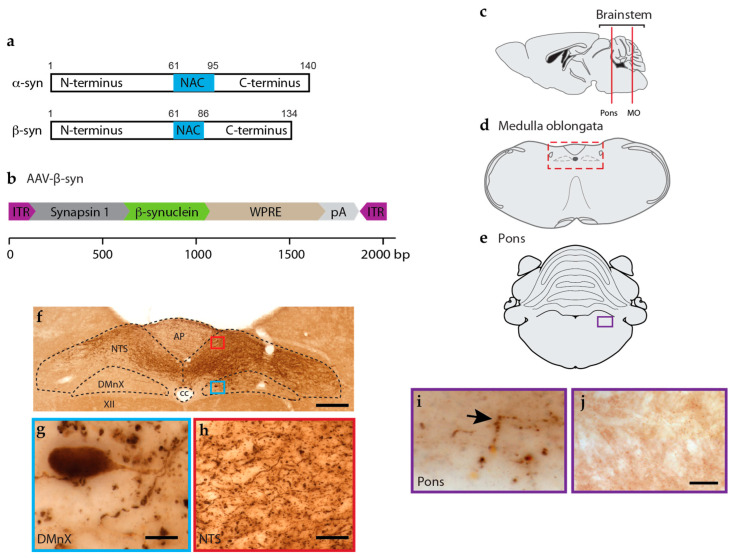
AAV-β-syn injections into the mouse vagus nerve induce increased expression of β-syn in the dMO and trigger its spreading to the pons. (**a**) Schematic illustration of α- and β-syn proteins displaying similarities in size and domain composition. (**b**) AAV-β-syn vector design illustrating the expression cassette flanking two ITRs. The scale displays the size (base pair, bp) of each genetic component. Mice received an injection of AAV-β-syn into the left vagus nerve. At 6 weeks post-surgery, coronal sections of the medulla oblongata and the pons were stained with a β-syn antibody. (**c**) Locations of the medulla oblongata and pontine coronal sections used for histological analyses are illustrated on a longitudinal plane. (**d**) Schematic image of a coronal section of the MO. The rectangular box (red) delineates an area of the dMO corresponding to the area shown in panel f. (**e**) Schematic image of a coronal section of the pons. The rectangular box (purple) delineates an area corresponding to the pontine area shown in panels i and j. (**f**–**h**) Representative image of the dMO showing β-syn immunoreactivity (**f**). Dotted lines in (f) delineate the borders between AP, NTS, DMnX, XII and cc. The blue and red boxes mark the locations of the areas shown at higher magnification in (**g**,**h**). These higher magnification images were acquired from the DMnX (**g**) and NTS (**h**) ipsilateral to the injection side. Scale bars: 200 μm (**f**) and 20 μm (**g**,**h**). (**i**) β-syn immunoreactivity in a representative pontine section displaying β-syn-filled axons (arrow). Scale bar: 10 μm. (**j**) β-syn immunoreactivity in a pontine section collected from an untreated naïve mouse. Scale bar: 10 μm. Abbreviations: XII: nucleus of the hypoglossal nerve; AP: area postrema; cc: central canal; dMO: dorsal medulla oblongata; DMnX: dorsal motor nucleus of the vagus nerve; ITR: inverted terminal repeat; NAC: non-amyloid component; NTS: nucleus tractus solitarius; pA: polyA; WPRE: woodchuck hepatitis virus posttranscriptional regulatory element.

**Figure 2 cells-12-00569-f002:**
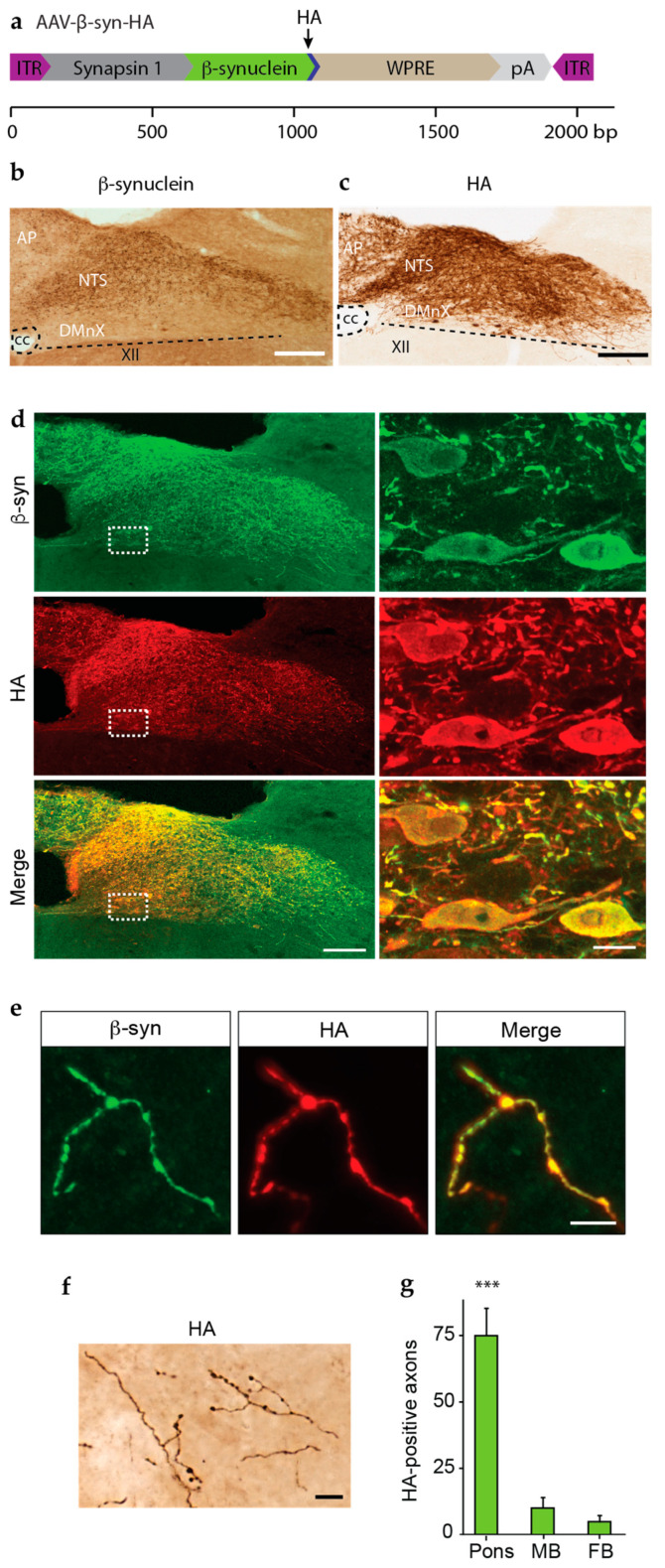
HA tagging allows sensitive and specific detection of β-syn protein. (**a**) Vector design of AAV-β-syn-HA. The scale bar displays the size (base pair, bp) of each genetic component. (**b**–**g**) Mice received an injection of AAV-β-syn-HA into the left vagus nerve and were sacrificed 6 weeks post-surgery. Brain tissue was processed for histological analyses. (**b** and **c**) Coronal sections of the medulla oblongata were immunostained with an antibody against β-syn (**b**) or HA (**c**). (**d**) β-syn (green) and HA (red) double immunolabeling in coronal sections of the dMO visualized by confocal microscopy. Panels on the right show transduced neuronal cell bodies in the DMnX at high magnification. Scale bars: 100 μm (low magnification) and 10 μm (higher magnification). (**e**,**f**) Coronal sections of the pons were either double-labeled with anti-β-syn (green) and anti-HA (red) and imaged with confocal microscopy (**e**) or stained with anti-HA and processed for brightfield microscopy (**f**). Scale bars: 10 μm (**e**) and 5 µm (**f**). (**g**) Counts of HA-positive axons in predefined coronal sections of the pons, MB, and FB. *F*_2,21_ = 38.56, *** *p <* 0.0005. Abbreviations: XII: nucleus of the hypoglossal nerve; AP: area postrema; cc: central canal; DMnX: dorsal motor nucleus of the vagus nerve; ITR: inverted terminal repeat; NTS: nucleus tractus solitarius; pA: polyA; WPRE: woodchuck hepatitis virus posttranscriptional regulatory element.

**Figure 3 cells-12-00569-f003:**
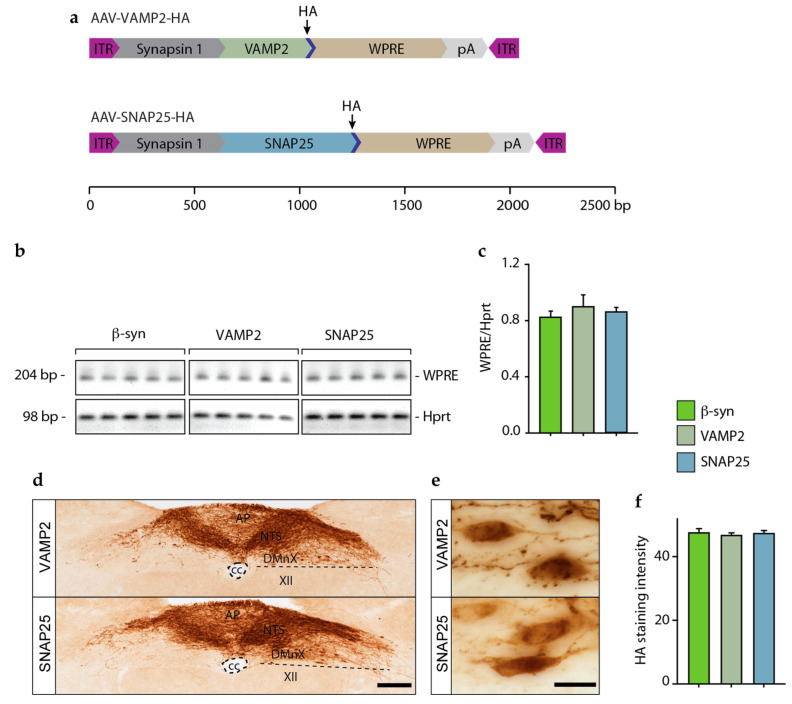
AAV-induced expression of β-syn, VAMP2, and SNAP25 in the dMO. Mice received an intravagal injection of AAV-β-syn-HA, AAV-VAMP2-HA, or AAV-SNAP25-HA and were sacrificed 6 weeks after surgery. (**a**) Vector design of AAV-VAMP2-HA and AAV-SNAP25-HA. The scale bar displays the size (base pair, bp) of each genetic component. (**b**,**c**) Tissue samples of the dMO ipsilateral to the injection site were collected from mice treated with AAV-β-syn-HA, AAV-VAMP2-HA, or AAV-SNAP25-HA (n = 5/group). RT-PCR was performed to detect WPRE and the housekeeping gene Hprt. Specific bands were detected at 204 (WPRE) and 98 (Hprt) bp (**b**). Densitometric measurements of WPRE and Hprt band intensities were plotted as WPRE to Hprt ratio. ANOVA, *F*_2,12_ = 0.46, *p =* 0.64 (**b**). (**d**,**e**) Representative images show medulla oblongata sections from mice injected with AAV-VAMP2-HA or AAV-SNAP25-HA that were stained with anti-HA (**d**). High magnification images show robust VAMP2 or SNAP25 expression within cell bodies of the DMnX (**e**). Scale bars: 200 μm (**d**) and 20 μm (**e**). (**f**) Intensity of HA labeling (arbitrary units) in the dMO ipsilateral to the side of AAV injection ANOVA, *F*_2,12_ = 0.23, *p* = 0.80. Abbreviations: XII: nucleus of the hypoglossal nerve; AP: area postrema; cc: central canal; DMnX: dorsal motor nucleus of the vagus nerve; ITR: inverted terminal repeat; NTS: nucleus tractus solitarius; pA: polyA; WPRE: woodchuck hepatitis virus posttranscriptional regulatory element.

**Figure 4 cells-12-00569-f004:**
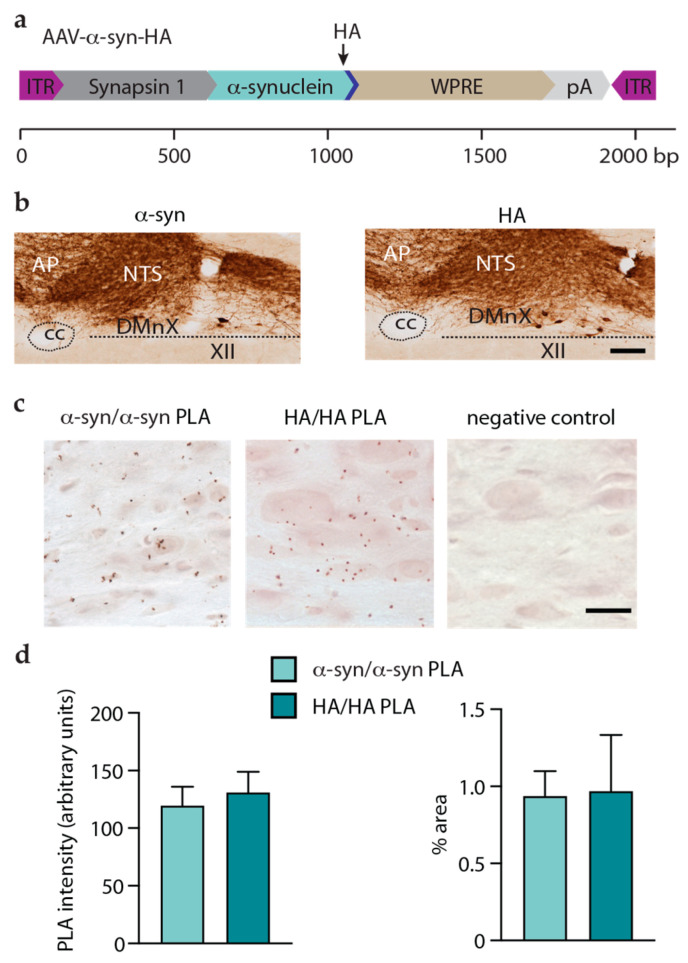
AAV-induced expression and aggregation of α-syn in the dMO. Mice received an injection of AAV-α-syn-HA into the left vagus nerve and were sacrificed 6 weeks after surgery. (**a**) Vector design of AAV-α-syn-HA. The scale bar displays the size (base pair, bp) of each genetic component. (**b)** Representative images of coronal sections of the medulla oblongata that were stained either with an antibody specific to human α-syn or with anti-HA. Scale bar: 100 μm. (**c**) Coronal sections of the medulla oblongata were processed for either α-syn/α-syn or HA/HA PLA. Representative images show specific PLA labeling (reddish brown) in the DMnX. No labeling was instead detected in negative control sections in which one of the HA/HA PLA probes was omitted. Scale bar: 10 μm. (**d**) Image analyses were carried out in the left (ipsilateral to AAV injections) dMO and measured the intensity of the PLA signal (left panel) and the percentage of the area occupied by this signal (right panel). Unpaired *t*-test, *t*_4_= 0.48, *p* = 0.90 (left panel) and *t*_4_ = 0.08, *p* = 0.32 (right panel). Abbreviations: XII: nucleus of the hypoglossal nerve; AP: area postrema; cc: central canal; DMnX: dorsal motor nucleus of the vagus nerve; ITR: inverted terminal repeat; NTS: nucleus tractus solitarius; pA: polyA; WPRE: woodchuck hepatitis virus posttranscriptional regulatory element.

**Figure 5 cells-12-00569-f005:**
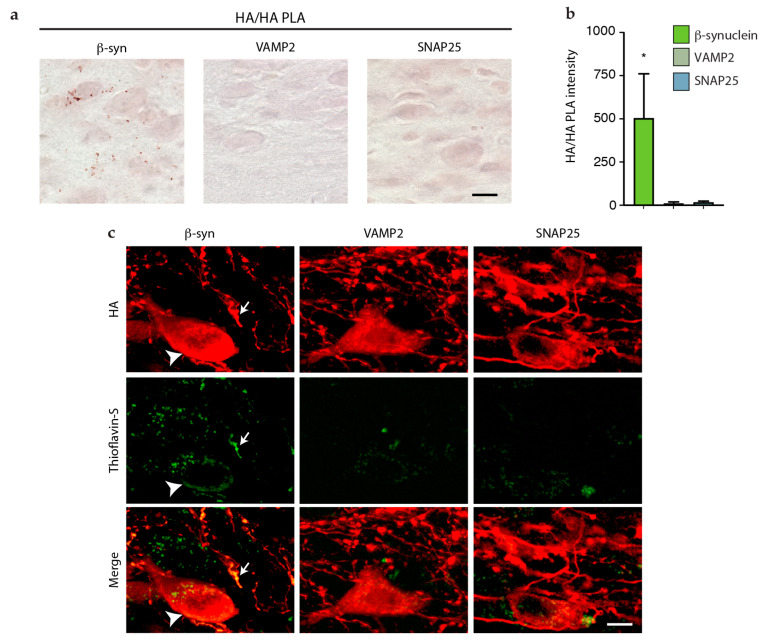
Aggregation of β-syn but not VAMP2 and SNAP25 in dMO. Mice received an injection of AAV-β-syn-HA, AAV-VAMP2-HA, or AAV-SNAP25-HA into the left vagus nerve and were sacrificed 6 weeks after surgery. Coronal sections of the medulla oblongata were processed for HA/HA PLA and HA/Thioflavin-S co-staining. (**a**) Representative images show specific PLA labeling (reddish brown) only in the DMnX of mice treated with AAV-β-syn-HA. Scale bar: 10 μm. (**b**) Image analyses were carried out in the left (ipsilateral to AAV injections) dMO and measured the intensity of the PLA signal. Data are expressed as arbitrary units. ANOVA, F_2,11_ = 4.79, * *p <* 0.05. (**c**) Representative images show Thioflavin-S labeling (green) within HA-immunoreactive neurites (arrow) and cell bodies (arrowhead) in the DMnX of mice treated with AAV-β-syn-HA but not AAV-VAMP2-HA or AAV-SNAP25-HA. Scale bar: 10 μm.

**Table 1 cells-12-00569-t001:** Counts of HA-positive axons in predefined sections of the pons, midbrain, and forebrain.

		Mean Number of HA-Immunoreactive Axons		
Treatment	Number of Samples	Pons (*F*_2,20_ = 50.24)	MB (*F*_2,20_ = 6.57)	FB (*F*_2,20_ = 5.64)
AAV-β-syn-HA	8	75 ± 10 ***	10 ± 4 *	5 ± 2 *
AAV-VAMP2-HA	8	0	0	0
AAV-SNAP25-HA	7	3±1	0	0

ANOVA followed by Tukey posthoc test. *** = *p <* 0.0001 when compared with AAV-VAMP2-HA and SNAP25-HA groups; * = *p <* 0.05 when compared with AAV-VAMP2-HA and SNAP25-HA groups.

## Data Availability

All data are available in the main text.

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
