# Peer review of "Interneuronal In Vivo Transfer of Synaptic Proteins"

_cells, 2023, doi:10.3390/cells12040569_

Round 1
Reviewer 1 Report
A central topic in Parkinson’s disease research concerns the phenomenon by which misfolded alpha-synuclein (a-Syn) proteins spread pathology from their initial site of deposition to other, more rostral, anatomically-connected sites in the brain. This is an important topic, and how specific this process is to the a-Syn protein is still not clear.
In this regards, Klinkenberg and co-workers set out to answer the question as to which features of the a-Syn protein might determine its interneuronal spreading potential; specifically, could this be related to the protein’s: (i) size, (ii) pre-synaptic localisation, and/or (iii) aggregated state. Thus, the authors appropriately choose 3 other proteins to investigate this question: the synuclein family protein beta-synuclein (related in [i], [ii], and [iii] to a-Syn), and the SNARE proteins VAMP2 and SNAP25 (related in [i] and [ii] to a-Syn).
The methodology involved a unilateral injection of female mice into the left vagus nerve with AAV vectors mediating overexpression of human b-Syn-HA, VAMP2-HA, and SNAP25-HA under control of a human synapsin promoter. Spread was assessed 6 weeks after surgery, when the mice were sacrificed. Use of the C-terminal HA tag allowed distinction of exogenous (human) b-Syn from its endogenous (murine) counterpart, and also enabled a quantitative comparison of protein spreading between the 3 proteins using the same anti-HA antibody. Importantly, the authors demonstrated comparable transduction and overexpression levels of the 3 proteins. The HA tag was also utilised to successfully validate a new HA/HA proximal ligation assay for detection of aggregated HA-tagged proteins. The overall quality of the methods employed and experimental design is highly satisfactory with use of appropriate controls.
The authors essentially show that the b-Syn protein, but not VAMP2 or SNAP25, is able to spread to the pons, midbrain and forebrain regions after 6 weeks, and b-Syn is seen within enlarged, dystrophic axons. Importantly, b-Syn is the only one of the 3 proteins to form aggregates. Hence, the authors conclude that the capacity of a protein for interneuronal spread correlates not with its size, neuronal or synaptic location, or mere overexpression, but critically with its ability to form protein aggregates. This is a significant and novel conclusion that contributes to advancement of the field, not only with respect to PD, but also for neurodegenerative diseases of the amyloid type more generally.
Statistical tests are applied correctly to the data, and sufficient experimental and statistical detail is given in the figure legends.
The conclusions are largely justified by the data, however the authors should respond to the following criticisms:
1. The HA-tag can be cleaved by caspase-3 and caspase-7, which are activated in apoptotic cells. However, did authors did not test immunohistochemically for apoptotic markers. The authors should take this into account when interpreting their results.
2. The authors use statistical tests for normally distributed data, but did they actually test the data for normality?
3. How many mice were used for the experiments?
4. It is interesting that b-Syn aggregated and manifested spreading, and supports other work indicating a neurodegenerative potential of b-Syn. However, can the authors explain why there is such a drastic drop in b-Syn positive axons from the pons (~75%) to the midbrain (~10%)? For instance, in an earlier paper on a-Syn spread (Ulusoy et al. 2013) the authors had found only 35% a-Syn positive axons in pons and 10% in the midbrain - and at 8 weeks. Did they look at later timepoints, e.g. >12 weeks?
5. The authors do not present data on spread of b-Syn to the side of the brain contralateral to the site of viral injection. Was this looked into?
6. Did the authors attempt Thioflavin-T staining of the b-Syn aggregates to check for fibril formation?
7. Did the authors check for spread to the SNc - an essential feature of PD?
Author Response
We thank the reviewer for her/his comments. We have prepared a revised version of the manuscript addressing these comments. Please find our point-by-point responses below:
- The HA-tag can be cleaved by caspase-3 and caspase-7, which are activated in apoptotic cells. However, did authors did not test immunohistochemically for apoptotic markers. The authors should take this into account when interpreting their results.
We agree that the HA tag may be cleaved in these experimental conditions, yet we do not think that our results will be significantly affected by this as we consistently detect strong HA expression and HA and B-syn colocalization in the donor neurons in the medulla (affected by AAV transduction) and in the recipient neurons (taking up the exogenous protein).
We also agree with the reviewer that this point could be stressed better and therefore cited a manuscript pointing out this possibility and added the following sentences in the discussion (see page 13, lines 433-440)
„It is noteworthy that cleavage and degradation of the HA tag, which have been reported by earlier investigations [39], might have hampered our assessment of protein overexpression (in the dMO) and protein spreading (in regions rostral to the medulla oblongata) using HA immunoreactivity. HA cleavage and clearance cannot be completely ruled out under our experimental conditions. Nonetheless, our findings do unequivocally indicate accumulation of b-syn-HA in the dMO and caudo-rostral spreading of b-syn-HA, since co-labeling analyses consistently detected co-localization of HA and b-syn immunoreactivities“
- The authors use statistical tests for normally distributed data, but did they actually test the data for normality?
We apologize for this missing information. We modified our materials and methods in the revised manuscript to clarify that the data is tested for normality (see page 5, lines 214-215).
„Prior to using parametric tests, normal distribution of the data was confirmed using Shapiro-Wilk test for normality.“
- How many mice were used for the experiments?
The size of the experimental groups was indicated as degrees of freedom in each statistical comparison as part of the F statistics. In this revised version we also added the number of animals used for each experiment in the materials and methods.
- It is interesting that b-Syn aggregated and manifested spreading, and supports other work indicating a neurodegenerative potential of b-Syn. However, can the authors explain why there is such a drastic drop in b-Syn positive axons from the pons (~75%) to the midbrain (~10%)? For instance, in an earlier paper on a-Syn spread (Ulusoy et al. 2013) the authors had found only 35% a-Syn positive axons in pons and 10% in the midbrain - and at 8 weeks. Did they look at later timepoints, e.g. >12 weeks?
This is an interesting point. Although one can speculate that this drop in the number of b-syn carrying fibers in the midbrain and forebrain may imply subtle differences in spreading behavior between a- and b-syn, this comparison would not be entirely sound as these previous experiments were performed using different viral titers and at different time points. Further experiments are required to address this point, which we believe that is beyond the scope of this manuscript.
- The authors do not present data on spread of b-Syn to the side of the brain contralateral to the site of viral injection. Was this looked into?
In the revised manuscript we analyzed the contralateral side of the brain for b-syn spreading and reported on page 8, lines 294-298. “
“The number of axons containing exogenous b-syn was consistently 60-70% lower on the contralateral as compared to the ipsilateral side of the brain. In particular, when bilateral counts were carried out in the pons of 5 AAV-injected mice, the number of b-syn-labeled axons was found to be 32.6 ± 6.1 and 87.6 ± 12.6 contralaterally and ipsilaterally, respectively.”
- Did the authors attempt Thioflavin-T staining of the b-Syn aggregates to check for fibril formation?
We have now performed Thioflavin-S staining in the dMO from mice expressing b-syn, VAMP2 and SNAP25. Tissue sections were also immunostained for HA to detect transduced neurons. Results are consistent with the results of HA/HA PLA analyses and show Thioflavin/HA co-localization in the medullary sections from mice injected with AAV-b-syn-HA but not in mice treated with AAV-VAMP2-HA or AAV-SNAP25-HA. New experiments are described in the materials and methods section (page 4, lines 174-182), results are presented on page 11, lines 388-393, and discussed on page 14, lines 491-493. The results are also illustrated as part of the new figure 5, panel c.
- Did the authors check for spread to the SNc - an essential feature of PD?
The fiber count analyses and the reports are performed throughout the brain and we have never observed spreading to the SNc in any of our experiments. This point is now mentioned in the results section on page 8, line 290 :
“…the pons, the reticular formation (but not the substantia nigra)…”
And discussed in the discussion section on page 12, lines 410-411:
“The substantia nigra was not among these regions, probably because of its weak direct connections to the DMnX [22, 38].“
Reviewer 2 Report
This study aims to address the nature of the trans-neuronal spread of synuclein proteins implicated in Parkinson’s disease. Previously, the authors demonstrated that alpha-synuclein but not GFP has the ability to spread along neuronal pathways after localized protein overexpression in the vagus nerve. Here the authors demonstrated that beta-synuclein, being highly homologous to the alpha isoform, follows the same behaviour. Overexpression of synuclein(s) targeted to the dorsal medulla oblongata (dMO) led to caudo-rostral neuron-to-neuron transfer towards pontine, midbrain, and forebrain regions. This is a very interesting observation that sheds additional light on protein behaviour in neurodegenerative diseases. The manuscript is written well, and all technical aspects are presented to a high standard. However, there is one caveat related to their choice of negative controls to address whether all synaptic proteins have the propensity to spread following a targeted overexpression. The VAMP protein is a transmembrane vesicular protein and thus it would not follow the same behaviour as cytosolic synucleins. SNAP-25 is a palmitoylated protein that is also strictly found on membranes, often in tight association with a transmembrane protein, syntaxin. Such SNARE properties would present obstacles to neuron-to-neuron transfer. A better choice for controls would be soluble synaptic proteins which may or may not aggregate. Although synucleins were shown to bind lipids and membranes (mostly in vitro), the bulk of cellular synucleins exists in the cytosol and this is where alpha-synuclein aggregates. Another slight issue is the choice of PLA as the only method to ascertain protein aggregation. Possible mutagenesis of synucleins resulting in, or preventing, the formation of protein aggregates would add greatly to their conclusions but this can be addressed in future studies. Finally, it would be good to mention in Introduction the neuronal distribution of two synucleins – do they co-exist in the same neurons or are they distributed in a non-overlapping manner?
Author Response
We thank the reviewer for her/his comments. We have prepared a revised version of the manuscript addressing these comments. Please find our point-by-point responses below:
However, there is one caveat related to their choice of negative controls to address whether all synaptic proteins have the propensity to spread following a targeted overexpression. The VAMP protein is a transmembrane vesicular protein and thus it would not follow the same behaviour as cytosolic synucleins. SNAP-25 is a palmitoylated protein that is also strictly found on membranes, often in tight association with a transmembrane protein, syntaxin. Such SNARE properties would present obstacles to neuron-to-neuron transfer. A better choice for controls would be soluble synaptic proteins which may or may not aggregate. Although synucleins were shown to bind lipids and membranes (mostly in vitro), the bulk of cellular synucleins exists in the cytosol and this is where alpha-synuclein aggregates.
We agree with the reviewer that synaptic proteins with different binding properties may behave differently. We acknowledged this point in our revised manuscript and discussed this in the discussion section on page 13, lines 449-453:
„These marked differences in spreading ability may reflect important features of these synaptic proteins, such as their relative solubility (synucleins) and their ability to stably interact with synaptic membranes (VAMP2 or SNAP25), that could facilitate or impede their spreading potential. Further studies are warranted to address this hypothesis.
Another slight issue is the choice of PLA as the only method to ascertain protein aggregation.
We have now performed Thioflavin-S labeling, a fluorescent dye that binds to amyloid structures, in the dMO from mice expressing b-syn, VAMP2 and SNAP25. Tissue sections were also immunostained for HA to detect transduced neurons. Results are consistent with the results of HA/HA PLA analyses and show Thioflavin-S/HA co-localization in the medullary sections from mice injected with AAV-b-syn-HA but not in mice treated with AAV-VAMP2-HA or AAV-SNAP25-HA. New experiments are described in the materials and methods section (page 4, lines 174-182), results are presented on page 11, lines 388-393, and discussed on page 14, lines 491-493. The results are also illustrated as part of the new figure 5, panel c.
Finally, it would be good to mention in Introduction the neuronal distribution of two synucleins – do they co-exist in the same neurons or are they distributed in a non-overlapping manner?
We have now added a new sentence and a citation to address this point on page 2, lines 88-89.
„b-syn is not only expressed within the same neuronal populations in the brain but also shares with a-syn its synaptic localization [28].“